# Ginger Oil Nanoemulsion Formulation Augments Its Antiproliferative Effect in Ehrlich Solid Tumor Model

**DOI:** 10.3390/foods12224139

**Published:** 2023-11-15

**Authors:** Danah S. Alharbi, Shouq F. Albalawi, Sarah T. Alghrid, Basma S. Alhwity, Mona Qushawy, Yasmin Mortagi, Mohamed El-Sherbiny, Kousalya Prabahar, Nehal Elsherbiny

**Affiliations:** 1Pharm D Program, Faculty of Pharmacy, University of Tabuk, Tabuk 71491, Saudi Arabia; danah.s.m.alharbi@gmail.com (D.S.A.); shouq.f.albalaawi@gmail.com (S.F.A.); sarah.alghrid@gmail.com (S.T.A.); basma.s19000@gmail.com (B.S.A.); 2Department of Pharmaceutics, Faculty of Pharmacy, University of Tabuk, Tabuk 71491, Saudi Arabia; 3Department of Pharmaceutics, Faculty of Pharmacy, Sinai University, Alarish 45511, North Sinai, Egypt; yasmin.mohamed@su.edu.eg; 4Department of Basic Medical Sciences, College of Medicine, AlMaarefa University, Riyadh 13713, Saudi Arabia; msharbini@um.edu.sa; 5Department of Anatomy and Embryology, Faculty of Medicine, Mansoura University, Mansoura 35516, Egypt; 6Department of Pharmacy Practice, Faculty of Pharmacy, University of Tabuk, Tabuk 71491, Saudi Arabia; kgopal@ut.edu.sa; 7Department of Pharmaceutical Chemistry, Faculty of Pharmacy, University of Tabuk, Tabuk 71491, Saudi Arabia; nelsherbiny@ut.edu.sa; 8Department of Biochemistry, Faculty of Pharmacy, Mansoura University, Mansoura 35516, Egypt

**Keywords:** ginger oil, nanoemulsion, Ehrlich’s ascites carcinoma, apoptosis, tumor, breast cancer

## Abstract

Cancer is a disease that is characterized by uncontrolled cell proliferation. Breast cancer is the most prevalent cancer among women. Ginger oil is a natural cancer fighter and anti-oxidant. However, the minimal absorption of ginger oil from the gastrointestinal tract accounts for its limited medicinal efficacy. The present study was designed to evaluate the efficacy of a nanoemulsion preparation of ginger oil on its oral bioavailability and in vivo anti-cancer efficacy. Ginger oil nanoemulsion was prepared by a high-pressure homogenization technique using different surfactants (Tween 20, 40, and 80). The prepared formulations were evaluated for droplet size, polydispersity index (PDI), zeta potential (ZP), pH, viscosity, and stability by calculating the creaming index percentage. The best formulation was evaluated for shape by TEM. The antitumor activity of the best nano-formulation was determined in comparison with the free oil using the in vivo Ehrlich solid tumor (EST) model. The prepared ginger oil nanoemulsion formulations exhibited acceptable droplet size in the range from 56.67 ± 3.10 nm to 357.17 ± 3.62 nm. A PDI of less than 0.5 indicates the homogeneity of size distribution. The oil globules possessed a negative charge ranging from −12.33 ± 1.01 to −39.33 ± 0.96 mV. The pH and viscosity were in the acceptable range. The TEM image of the best formulation appeared to be spherical with a small size. The ginger oil nanoemulsion reduced in vivo tumor volume and weight, extended animals’ life span, and ameliorated liver and kidney function in EST-bearing mice. These effects were superior to using free ginger oil. Collectively, the present study demonstrated that the ginger oil nanoemulsion improved oral absorption with a subsequent enhancement of its anti-proliferative efficacy in vivo, suggesting a nano-formulation of ginger oil for better therapeutic outcomes in breast cancer patients.

## 1. Introduction

Cancer is a disorder in which the proliferation of cells is uncontrollable [1]. Cancer is the second most common cause of death around the globe [2]. There are around 100 different types of cancer depending on which cell is afflicted [3]. Tumors can grow and interfere with the circulatory system, the central nervous system, and the digestive system in some situations, affecting the quality of life of cancer patients and leading to an increased mortality rate [4]. Breast cancer is the most diagnosed cancer among women with a steadily increased incidence [5].

Ehrlich’s ascites carcinoma (EAC) is an undifferentiated carcinoma with a fast rate of proliferation and a strong capacity to transplant [6]. It comes from a mouse model of breast cancer. It resembles human breast cancer with great anticancer medication sensitivity [7,8]. Many prior studies used Ehrlich’s solid tumor (EST) as an experimental model to assess medication and natural-ingredient anticancer activity [9,10]. Furthermore, because the EAC suspension contains homogeneous free tumor cells, it can be transplanted into another mouse in specified quantities [11]. Finally, the EAC cell line is simple to prepare, culture, and use in in vivo models.

Ginger, the rhizome of *Zingiber officinale* Roscoe, is one of the most widely used spices and a traditional remedy for pain, inflammation, and gastrointestinal problems [12]. Ginger essential oil is categorized as a warming essential oil, demonstrating antimicrobial, laxative, tonic, and stimulant effects. The health benefits attributed to ginger essential oil closely resemble the therapeutic benefits of fresh ginger. The efficacy of ginger essential oil is attributed to its high content in gingerol. Ginger oil is made from the fresh rhizomes of *Zingiber officinale* [13,14]. It has the same aroma and flavor as the spice, but it is not quite as strong [15]. Ginger essential oil has also been found to have antibacterial, antiviral, and antifungal properties [16]. Monoterpenes such as phellandrene, camphene, cineole, linalool, limonene, citral, geraniol, citronellol, borneol, and sesquiterpenes such as α-zingiberene, ar-curcumene, β-bisabolene, β-sesquiphellandrene, zingiberol, and zingiberenol are the main constituents of ginger oil [17]. It also contains anticancer compounds such as terpenoids, phenylpropanoids, flavonoids, and sesquiterpenes [18]. Bioactive chemicals included in this oil, such as 6-gingerol and zerumbone, have been shown to cause apoptosis in cancer cells [19]. As per the Food and Drug Administration (FDA), this oil is classified as “generally recognized as safe” (GRAS), thereby making it appropriate for utilization in food-related contexts [20]. Moreover, Kottarapat et al. reported the confirmed safety of ginger oil in male and female rats following subchronic oral administrations of up to 500 mg/kg per day [21].

Many researchers studied the effect of ginger against different types of cancer cells. Zaid et al. studied the effect of ginger oil on the human cervical cancer cell line, and they found that the oil demonstrated a strong antiproliferation potential [22]. Also, Zhang et al. concluded in their study that ginger produced preventive effects against colon cancer [23]. Ahmed et al. studied the effect of ginger extract on antioxidant status in an experimental model of liver cancer [24]. The researchers reached the conclusion that ginger extract potentially contains bioactive constituents that exhibit antioxidant properties by effectively scavenging free radicals, including superoxide anions and H_2_O_2_. Additionally, the extract was found to reduce the level of malondialdehyde, thereby mitigating lipid peroxidation. Bioactive constituents of ginger oil demonstrated an anti-proliferative effect in breast cancer in vivo [25] and in vitro [26].

Nanoemulsions are nanoscale emulsions that are utilized to increase the delivery of active therapeutic ingredients [27]. These are thermodynamically stable isotropic systems in which two immiscible liquids are combined into a single phase using an emulsifying agent such as surfactant and cosurfactant [28]. Emulsions with droplet sizes ranging from 20 to 500 nanometers are known as nanoemulsions [29]. Nanoemulsions are extensively used in the creation of medicinal formulations for topical, ocular, intravenous, and other routes of delivery [30]. In this study, the nanoemulsion preparations of ginger oil were prepared to enhance the oral bioavailability and hence the anticancer activity against the experimental model of breast cancer. This study included the use of three types of hydrophilic surfactants with different hydrophilic–lipophilic balance (HLB) values, Tween 20, Tween 40, and Tween 80. Tweens refer to a series of nonionic surfactants that are generated from sorbitan esters [31]. These substances exhibit solubility or dispersibility in water, although their organic and oil solubilities vary significantly. Oil-in-water emulsifiers find applications in several industries such as pharmaceuticals, cosmetics, and cleaning compounds [32]. Tween 80 is characterized by the presence of oleic acid as its fatty acid side-chain, while Tween 40 has palmitic acid and Tween 20 contains lauric fatty acid [33,34]. The observed variation in these surfactants can be related to the hydrophilic–lipophilic balance (HLB) value. The hydrophilic–lipophilic balance (HLB) values of these surfactants are 16.7, 15.6, and 15 for Tween 20, Tween 40, and Tween 80, respectively. Propylene glycol was used as cosurfactant, which helped in the stabilization of the nanoemulsion and in the reduction in droplet size.

## 2. Materials and Methods

### 2.1. Materials

Ginger oil was purchased from EL-Captain Company (Cairo, Egypt). Tween 20, 40, and 80 were purchased from Sigma Chemical Company (Taufkirchen, Germany). Propylene glycol was obtained from Sigma-Aldrich (St. Louis, MO, USA). All other chemicals were of analytical grade.

### 2.2. Design and Preparation of Ginger Oil Nanoemulsion Using 3^2 Full Factorial Design

The design of ginger oil nanoemulsion formulation was based on 3^2 full factorial design using Design Expert version 11 (Stat-Ease, Minneapolis, MN, USA). Two formulation factors (independent variables) were employed at three levels, namely X1, type of surfactant (Tween 20, 40, and 80), and X2, concentration of surfactant and cosurfactant (10, 20, and 30%). The effect of independent variables in the dependent variables (responses) were studied, wherein the responses included Y1; droplet size, Y2; polydispersity index (PDI), and Y3; Zeta potential (ZP). Table 1 represents the independent and dependent variables.

A total of nine ginger oil nanoemulsions (O/W) were synthesized using the high-pressure homogenization process [35]. The accurate quantities of surfactant and cosurfactant were dissolved in a precise volume of water within a small beaker. The accurate amount of oil was incrementally introduced into the aqueous phase using a high-shear homogenizer (Heidolph, Schwabach, Germany) operating at 20,000 rpm. This process continued until the whole volume of oil was injected, after which the mixture was subjected to homogenization for an additional 10 min. The nanoemulsion formulations were held at a temperature of 5 °C for a duration of 24 h before conducting further investigations.

### 2.3. Determination of the Droplet Size of Ginger Oil in O/W Nanoemulsion, Polydispersity Index, and Zeta Potential

The droplet size, polydispersity index (PDI), and zeta potential (ZP) of all ginger oil nanoemulsion formulations that were generated were assessed using the dynamic light scattering approach using the Zetasizer instrument (Malvern Instruments Ltd., Malvern, UK). To determine the dimensions of the droplets and the zeta potential, the specimens of each formulation were diluted with distilled water in a ratio of 1:100 at a temperature of 25 °C. The determination of the zeta potential of the nanoemulsion was conducted by assessing the electrophoretic mobility of the oil droplets [36]. All measurements were conducted in triplicate.

### 2.4. pH Evaluation of the Ginger Oil Nanoemulsion

The pH of the nanoemulsion formulations was determined by employing a pH meter (JENWAY, Staffordshire, UK) after homogenization with water in a ratio of 1:9. The measurements were conducted in triplicate under ambient conditions [37].

### 2.5. Viscosity Evaluation

The viscosity of the nanoemulsion was measured by the Ostwald viscometer performed at a temperature of 25 ± 0.5 °C. The experiments were performed in triplicate [38,39].

### 2.6. Stability Test of Nanoemulsions

#### 2.6.1. Heating Cooling Cycle (Accelerated Stability Study)

Approximately 50 mL of each nanoemulsion formulation was placed into a glass bottle well sealed with the cap (*n* = 3), and then subjected to accelerated conditions (4 °C and 45 °C for 6 cycles) [40,41]. The formulations underwent six cycles of temperature fluctuation between 4 °C and 45 °C, with each temperature being maintained for a storage period exceeding 48 h [41]. By the end of this test, the nanoemulsions were evaluated for physical stability characterized by the percentage of creaming and cracking. The cracking of nanoemulsion occurs when the oil and water are separated and will not recombine. The creaming index (%CI) was calculated as follows [42,43]:%CI = (CC/CT) × 100
where CC = total height of cream layer and CT = total height of nanoemulsion.

#### 2.6.2. Centrifugation Test

The nanoemulsion formulations were subjected to centrifugation using a centrifuge (Biofuge, Primo Heraeus, Osterode, Germany) at a speed of 5000 rpm for 30 min to assess the occurrence of creaming or cracking inside the system. The system was visually evaluated to assess its appearance [44,45].

### 2.7. The Selection of the Best Nanoemulsion Formulation

The best formulation was chosen to complete the in vivo study based on minimizing the droplet size and PDI and maximizing the absolute value of ZP.

### 2.8. Transmission Electron Microscopy Image of Best Ginger Oil Nanoemulsion

The morphological analysis of emulsions was conducted using transmission electron microscopy (TEM). The nanoemulsions were diluted in a ratio of 1:100, resulting in an oil phase concentration of 1% (*v*/*v*). The preparation of specimens for transmission electron microscopy (TEM) viewing involved the combination of samples with a single droplet of a uranyl acetate solution with a concentration of 2% (*w*/*v*). Subsequently, the samples were adsorbed onto copper grids coated with 200 mesh formvar, followed by a drying process. The examination of the samples was conducted using a transmission electron microscope (JEOL^®^, Tokyo, Japan) [46].

### 2.9. Experimental Protocol

#### 2.9.1. Induction of Ehrlich Solid Tumor (EST)

This study was conducted following the ethical guidelines for investigations in laboratory animals and the experimental design was approved by the scientific research ethics committee at the Faculty of Pharmacy, Sinai University, Arish, Egypt (approval number SU-SREC-3-05-23). Female Swiss Albino mice (22–30 g weight) were left to acclimatize for one week with free access to food and water under standard laboratory conditions. Thereafter, animals were randomly allocated into the following four groups (*n* = 9):

Group I (normal): received vehicle and served as normal control for 21 days.

Groups II (EST): animals bearing EST and treated with vehicle for 21 consecutive days.

Group III (EST + Ginger oil): animals bearing EST and treated with free ginger oil at a dose of (100 mg/kg/day, orally) for 21 consecutive days.

Group IV: animals bearing EST and treated with ginger oil nanoemulsion at a dose of (100 mg/kg/day, orally) for 21 consecutive days.

EST was induced in groups II-IV by subcutaneous inoculation of Ehrlich Ascites Carcinoma (EAC) cells in the right thigh (5 × 10^5^ viable EAC cells in 0.1 mL/mice). Day zero of tumor implantation was assigned when the primary tumor size reached 50–100 mm^3^.

The volume of the tumor was measured every five days for a period of 21 days using a digital caliper and applying the formula A × B^2^ × 0.5, where A is the largest diameter and B is its perpendicular [47].

#### 2.9.2. Sample Collection

At the end of the experimental procedure, animals were weighed and then sacrificed under anesthesia using thiopental sodium. Tumors were excised and weighed. Three animals from each group were left for calculation of mean survival time. Blood was withdrawn by cardiac puncture. Blood samples were centrifuged to separate serum for further biochemical analyses. Tumor specimens were fixed in buffered formalin for further histopathological analyses.

#### 2.9.3. Mean Survival Time and Percentage Increase in Life Span

The following formulas were used for the calculation of mean survival time (MST) and percentage increase in life span (%ILS) [47].
MST = Sum of survival time for each mouse in a group (days)/total number of mice in the group
%ILS = (MST of treated group (Group III or Group IV)/MST of EST group) × 100

#### 2.9.4. Biochemical Assessment of Kidney and Liver Function

Serum samples were analyzed for creatinine and BUN as markers for kidney function and for alanine aminotransferase (ALT) and aspartate aminotransferase (AST) as markers of liver function using commercially available kits according to manufacturers’ instructions.

#### 2.9.5. Histopathology

Fixed tumor specimens were processed to be embedded in paraffin. Sections with 5 μm thickness were cut and then stained with hematoxylin and eosin (H&E). The slides were inspected for morphological changes and photographed using a camera-aided light microscope (Olympus, Tokyo, Japan).

#### 2.9.6. Statistical Analysis

Data were expressed as mean ± standard error of the mean (mean ± SE). One-way analysis of variance (ANOVA) was used to assess the statistical differences between different group followed by Tukey’s post hoc analysis. Difference was considered significant among experimental groups when *p* value was less than 0.05. GraphPad prism software Version 8, was used for statistical analysis and graphical presentation.

## 3. Results and Discussion

According to 3^2 full factorial design, nine formulations were designed with different compositions, as represented in Table 2. All formulations were prepared by high shear homogenization technique and evaluated for the droplet size, PDI, ZP, pH, viscosity, and stability.

### 3.1. Effect of Formulation Factors on the Droplet Size of Ginger Oil Nanoemulsion

The droplet size of the prepared ginger oil nanoemulsion was determined by the dynamic light scattering technique. The results represented in Table 3 show that all formulations exhibited a small droplet size in the nano-range (less than 500 nm). The droplet size ranged from 56.67 ± 3.10 nm for F9 to 357.17 ± 3.62 nm for F1. As shown in Figure 1 and Table 4, it was found that the droplet size was significantly decreased (*p* < 0.05) using Tween 80 rather than Tween 40 and Tween 20. These results may be attributed to the difference in HLB of the used surfactants, as the HLB values of Tween 20, Tween 40, and Tween 80 were 16.7, 15.6, and 15 respectively. It can be argued that the utilization of surfactants with greater hydrophilic–lipophilic balance (HLB) values may lead to the production of nanoemulsions characterized by larger droplet sizes. A decrease in the hydrophilic–lipophilic balance (HLB) value corresponds to an increase in the surfactant’s affinity for the oily phase. This increased affinity leads to a subsequent decrease in interfacial tension and, consequently, a reduction in droplet size. Furthermore, it has been documented that the molecular geometry of surfactants plays a role in the formation of smaller sizes, particularly in the case of surfactants exhibiting identical polarity. The utilization of Tween 80 surfactant has been documented to result in the formation of smaller droplets inside the nanoemulsion system, in contrast to formulations including Tween 20 and Tween 40. Nevertheless, due to the distinctive molecular geometry of the Tween 80 surfactant, which includes unsaturated hydrocarbon segments and exhibits increased kinking, it has the potential to influence the arrangement of surfactant molecules at the oil–water interface. Consequently, this can result in the propensity for the creation of smaller droplets. The oily phase is combined with the aqueous phase in a natural manner [48].

Additionally, the droplet size was significantly decreased (*p* < 0.0001) as the total concentration of surfactant and cosurfactant increased from 10% to 30%, as seen in Table 4. The observed result can be attributed to the reduction in interfacial tension, which occurs when the concentration of both surfactant and cosurfactant increases. This reduction leads to the formation of a small oil droplet within the internal phase of the emulsion [49]. Furthermore, the increase in the concentration of surfactant and cosurfactant results in an elevation of steric stability, hence impeding the droplet aggregation of a produced nanoemulsion. Noori et al. conducted a study in which they prepared a ginger oil nanoemulsion so as to enhance its antimicrobial and antioxidant effect, and they found that the small droplet size was attributed to Tween 80 [50].

The homogeneity in size distribution is measured by the value of PDI; the lower the value of PDI, the higher the homogeneity. As represented in Table 3, it was found that the PDI value of all ginger oil nanoemulsions was less than 0.5, which indicates the homogeneity of droplet size distribution. The previous finding was in agreement with Qushawy et al., who found that the PDI value of the prepared hemp seed oil nanoemulsion was less than 0.5 [37]. As shown in Figure 2 and Table 4, it was observed that the PDI value was decreased significantly (*p* < 0.05) as the concentration of surfactant and cosurfactant increased. These outcomes may be attributed to the increase in the steric stability of the nanoemulsion and result in a lesser internal phase aggregation tendency.

The surface charge at the interface of the droplets is determined by the zeta potential, which is influenced by the charge of surfactants adsorbed around the droplets. These surfactants may have anionic, cationic, or non-ionic characteristics. The nanoemulsions generated in this work were formulated with a non-ionic surfactant, which may result in an anticipated electrical charge proximate to zero. Nevertheless, all the produced nanoemulsion formulations exhibited a negative ZP. The prepared ginger oil nanoemulsion formulations exhibited a wide range of ZP values, ranging from −12.33 ± 1.01 to −39.33 mV, as seen in Table 3. The negative charge may be attributed to the existence of ionizable groups present in the ginger oil. When emulsions undergo intense mechanical forces, such as ultrasonication, this can result in the liberation of hydroxyl and carboxyl groups from the chemical structure of ginger oil. These groups then migrate toward the surface of the nanoemulsion droplets, where they become accessible for binding with water [50]. The deprotonated forms of alcohols (R-O-) and carboxylic acids (R-COO-) serve to augment the negative charge within the interfacial region of the droplets subsequent to the ultrasonication procedure. In a study conducted by Acevedo-Fani et al. (2015), it was shown that Tween 80 emulsified nanoemulsions containing sage essential oil exhibited a negative charge on their surface [51]. Figure 3 and Table 4 show that the negative charge increased significantly (*p* < 0.05) using the smallest concentration of surfactant and cosurfactant (10%). These findings may be attributed to the fact that the increase in concentration of the non-ionic surfactant may neutralize and decrease the negative charge on the surface of the oil droplets.

### 3.2. pH Measurement of Ginger Oil Nanoemulsion

The pH value of the ginger oil nanoemulsion formulations was assessed. According to the data reported in Table 5, the pH values varied between 5.87 ± 0.03 for F6 and 6.87 ± 0.02 for F9. A comparable result was achieved by Ningsih et al., who formulated a nanoemulsion of ginger oil in order to augment its antioxidant efficacy [52]. Furthermore, Sondari et al. conducted a study in which they formulated a nanoemulsion of ginger oil using surfactants Span 80 and Tween 85. The researchers observed that the pH value of the resulting formulation ranged from 6.65 ± 0.011 to 6.72 ± 0.012 [53].

### 3.3. Viscosity of Ginger Oil Nanoemulsion

The viscosity of the nanoemulsion is one of the important physical parameters which should be measured. As represented in Table 5, it was found that the viscosity of the ginger oil nanoemulsion formulation ranged from 2.31 ± 0.03 cP for F7 to 20.25 ± 0.08 cP for F3.

The viscosity of nanoemulsions exhibits a clear correlation with the concentration of surfactants for each specific type of surfactant. In addition, it should be noted that the viscosity value of nanoemulsions is influenced by the types of surfactants employed. Specifically, it has been shown that the formulation including Tween 20 exhibits the highest viscosity value, followed by Tween 40 and Tween 80. Aziz et al. conducted a study in which they prepared a nanoemulsion of eucalyptus oil using different concentrations (3.0, 6.0, 9.0, 12.0, 15.0, and 18.0 wt.%) of Tween 40, 60, and 80 as surfactants. The researchers found that the type of surfactant used had an impact on the viscosity of the nanoemulsions. Specifically, the formulation containing Tween 40 exhibited the highest viscosity value (5.45 cP), followed by Tween 80 (5.25 cP) and Tween 60 (5.09 cP) [54]. The increased viscosity of the formulation may be attributed to the formation of hydrogen bonds between the hydrophilic segments of the surfactant and water molecules. These hydrogen bonds result in the entrapment of water molecules within the cross-linking regions of the surfactant, leading to a significant increase in viscosity. The increase in viscosity of the micellar formulation can be attributed to a higher concentration of surfactant cross-linking sections [55]. Furthermore, it is worth noting that the primary factor contributing to the elevated viscosity value of Tween 20 nanoemulsions is its comparatively higher hydrophilic–lipophilic balance (HLB) value in comparison to the other two surfactants. Surfactants having a high hydrophilic–lipophilic balance (HLB) possess a greater proportion of hydrophilic components, hence facilitating an increased interaction with water molecules inside nanoemulsion systems [56].

### 3.4. Stability Study of Ginger Oil Nanoemulsion

The evaluation of kinetic stability is of great importance since it not only provides insights into the immediate stability of nanoemulsions but also allows for predictions regarding their long-term stability. The augmentation of temperatures and the utilization of centrifugal forces contribute to the amplification of Brownian motion, hence helping the convergence of dispersed droplets [41].

The type of surfactant employed exerts a discernible influence on the properties of emulsions. The creaming index percentage was shown to be significantly influenced by the type of surfactants, as seen in Table 5. It was found that the CI% ranged from 1.61 ± 0.25% for F9 to 21.24 ± 0.60% for F1. According to the results, it was observed that the formulation developed in the presence of Tween 80 had a reduced creaming index, suggesting enhanced physical stability. The observed outcomes can be ascribed to the disparity in the hydrophilic–lipophilic balance (HLB) values of the surfactants employed. Moreover, an increase in the concentration of both surfactant and cosurfactant leads to a decrease in the creaming index. The observed results can be ascribed to the decrease in interfacial tension and enhancement of stearic stability in the nanoemulsion. These outcomes are in agreement with the results obtained by Pengon et al., who found that the type of surfactant had a great impact on the stability of their coconut oil nanoemulsion [57].

The outcomes of the centrifugation test indicate that all nanoemulsions exhibited stability, as evidenced by the absence of phase separation or creaming subsequent to centrifugation. Similar results were obtained by Arianto and Cindy, who prepared a sunflower nanoemulsion and found that there was no creaming or cracking after centrifugation at 3750 rpm for 5 h [58].

### 3.5. Selection of the Optimized Formulation

The utilization of Design–Expert software Version 11, facilitated the selection of an optimum formulation by prioritizing the minimization of droplet size, polydispersity index (PDI), and maximization of negative zeta potential (ZP). Based on the optimization procedure, it was determined that F9 was chosen as the optimum formulation. This formulation was made utilizing Tween 80 as a surfactant, with a concentration of 30% for both the surfactant and cosurfactant in combination. According to the findings presented in Figure 4, it was observed that the anticipated values of the responses closely approximated the actual values, as indicated by a desirability index of 0.721.

### 3.6. Transmission Electron Microscopy (TEM) of Ginger Oil Nanoemulsion

The dimensions and shape of the nanoemulsion droplets containing ginger oil were determined using transmission electron microscopy (TEM) analysis. The transmission electron microscopy (TEM) image of the optimized formulation F9 revealed that the nanoemulsion of ginger oil had a spherical morphology, as seen in Figure 5. The observations revealed that the droplets had the characteristic appearance of an oil-in-water (O/W) emulsion. The increased contrast observed at the interface of the oil droplets can be attributed to the affinity of uranyl acetate, which is utilized as a negative staining agent, toward the components present at the interface. The obtained results are in full agreement with the results obtained by Shehabeldine et al., who prepared a clove oil nanoemulsion so as to enhance its antimicrobial and anticancer activity, and they found that the TEM image of the prepared clove oil nanoemulsion was spherical in shape [59].

### 3.7. Experimental Study of Ginger Oil and Ginger Oil Nanoemulsion on EST

As shown in Figure 6, the EST group showed a progressive increase in tumor volume reaching 1472.76 ± 82.16 mm^3^ on day 21. Tumor volume was reduced in the EST group treated with free ginger oil reaching 717.55 ± 45.21 mm^3^ on day 21. However, ginger oil nanoemulsion administration resulted in a more significant reduction in tumor volume, reaching 565.33 ± 22.74 mm^3^ on day 21. Consistently, tumor weight was decreased in free ginger oil and ginger oil nanoemulsion-treated groups (3.67 ± 0.14 gm and 2.4 ± 0.13 gm, respectively) when compared to the EST-untreated group (4.703 ± 0.065 gm), as seen in Figure 7. Decreased tumor volume and weight with treatment reflect the anti-proliferative efficacy of ginger oil, which was further improved by the nano preparation. Similarly, previous studies reported the antiproliferative effect of ginger oil on various types of cancer [22,60,61].

In addition to the effect of treatments on tumor volume and weight, histopathological data demonstrated that ginger oil caused shrinkage in tumor cells, increased necrotic area, and reduced viable cells. Of note, these effects were more favorable with ginger oil nanoemulsion treatment, as seen in Figure 8.

Table 6 demonstrates the effect of the treatments on the survival time and life span of mice bearing the Ehrlich solid tumor. Free ginger oil non-significantly increased the survival of EST-bearing mice when compared to the EST-untreated group (*p* = 0.14). However, the ginger oil nanoemulsion resulted in a significant increase in the survival of mice compared to both the EST-untreated group and the EST + free ginger oil-treated group (*p* < 0.01, *p* < 0.05, respectively).

Regarding kidney and liver functions, EST-bearing mice demonstrated deteriorated liver and kidney function, as illustrated by a significant increase in ALT, AST, creatinine, and BUN (*p* < 0.001, *p* < 0.0001, *p* < 0.0001, *p* < 0.001, respectively) when compared to the control group, as seen in Table 7. Treatment with free ginger oil resulted in a non-significant reduction in markers of liver and kidney function. However, treatment with ginger oil nanoemulsion resulted in a significant reduction in markers of liver and kidney function when compared to the EST-untreated group (*p* < 0.05, *p* < 0.01, *p* < 0.001, *p* < 0.05, respectively).

In addition to their anti-proliferative effect on EST, treatments with ginger oil and ginger oil nanoemulsion extended the life span of EST mice, which represents an important criterion for anticancer therapy. This effect on life span could be in part explained by the ameliorative effect of the treatments on kidney and liver function tests. Consistently, ginger oil extracts have been reported to provide hepato- and renal protection against many insults by various mechanisms including combating oxidative stress, inflammation, and apoptosis [62,63], encouraging its use as an adjuvant in radiotherapy and chemotherapy regime for the treatment of cancer or its implementation to counteract organ toxicities induced by alcohol, industrial pollutants, smoking, or administered drugs.

## 4. Conclusions

Ginger oil has the potential to be effectively formulated as a nanoemulsion. Through the implementation of a 3^2 full factorial design, the authors were able to achieve an optimized formulation characterized by a small droplet size (56.67 ± 3.10 nm), a low polydispersity index (0.340 ± 0.03), and a strong negative zeta potential (−29.20 ± 0.53 mV). The TEM image of the optimal formulation exhibited a spherical morphology characterized by a diminutive size. The use of ginger oil nanoemulsion resulted in a reduction in tumor volume and weight, an extension of the life span of mice with EST, and an improvement in liver and kidney function. The observed effects demonstrated a higher level of efficacy compared to the utilization of freely available ginger oil. Further studies are warranted to study the effect of prepared ginger oil nanoemulsion on other types of cancer. Also, its potential synergistic effect with chemotherapy needs further investigation.

## Figures and Tables

**Figure 1 foods-12-04139-f001:**
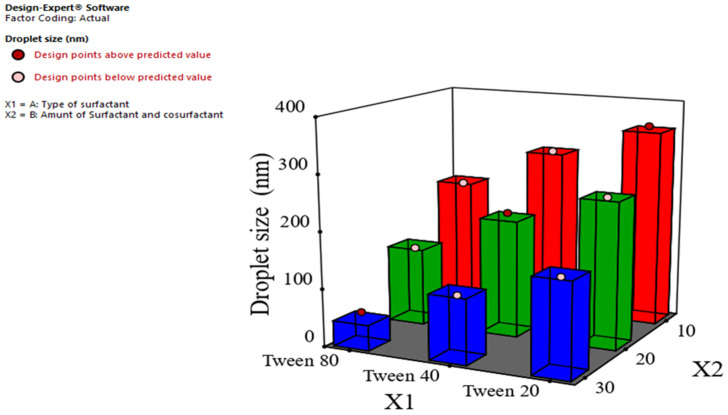
Effect of type of surfactant (X1) and amount of surfactant and cosurfactant (X2) in the droplet size (Y1).

**Figure 2 foods-12-04139-f002:**
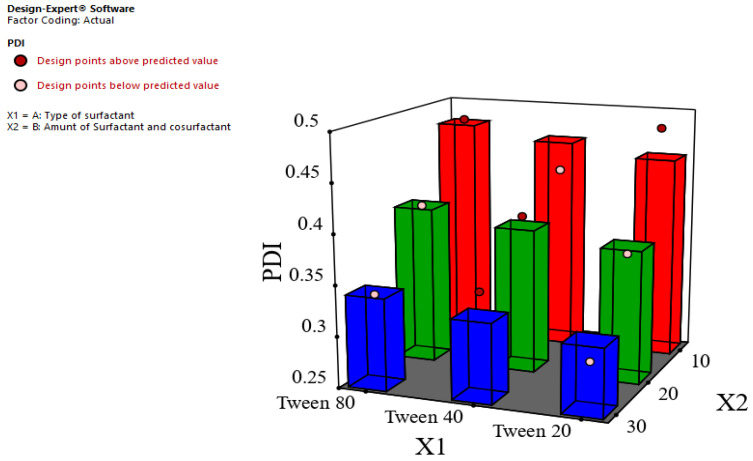
Effect of type of surfactant (X1) and amount of surfactant and cosurfactant (X2) on the PDI (Y2).

**Figure 3 foods-12-04139-f003:**
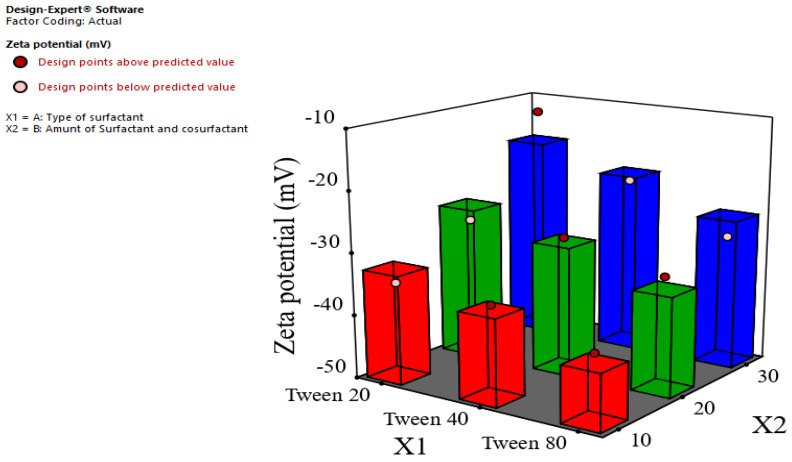
Effect of type of surfactant (X1) and amount of surfactant and cosurfactant (X2) on ZP (Y3).

**Figure 4 foods-12-04139-f004:**
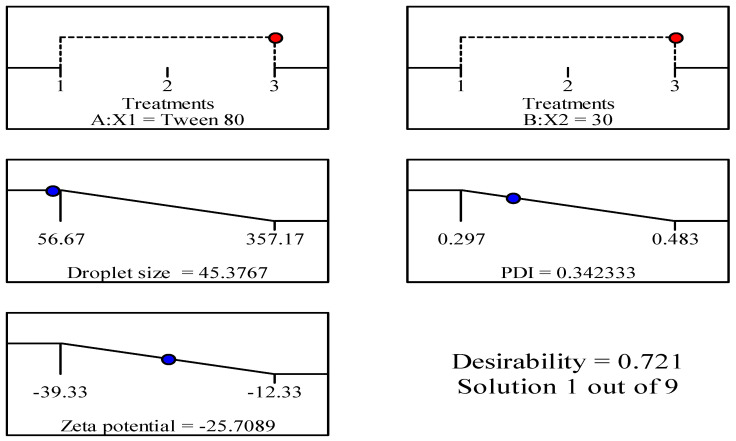
The composition of optimized formulation and the predicted values of responses.

**Figure 5 foods-12-04139-f005:**
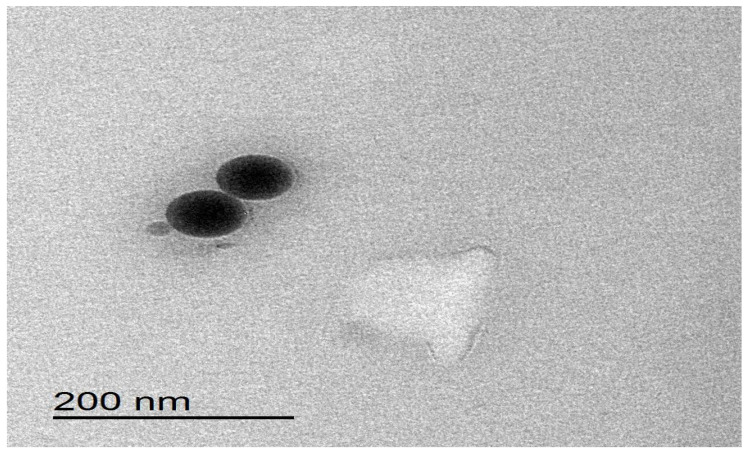
The TEM image of the optimized ginger oil nanoemulsion formulation F9.

**Figure 6 foods-12-04139-f006:**
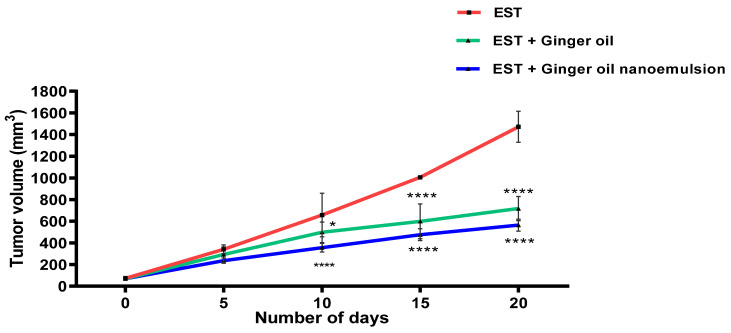
Time course effects of free ginger oil and ginger oil nanoemulsion on the growth of Ehrlich solid tumor (EST) in mice. Data are presented as mean ± SE of the solid tumor volume (mm^3^). * *p* < 0.05, **** *p* < 0.0001 compared to EST group.

**Figure 7 foods-12-04139-f007:**
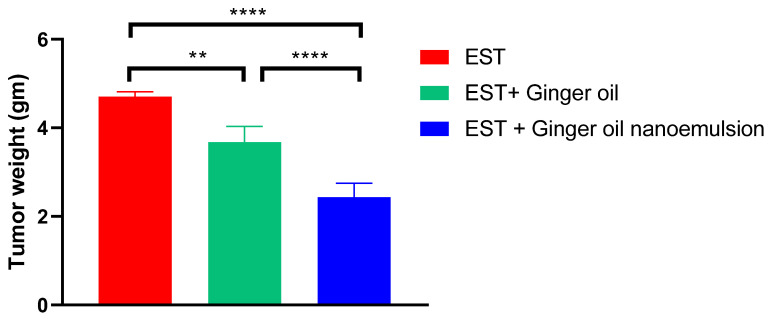
Effect of treatment with free ginger oil and ginger oil nanoemulsion on tumor weight. Data are presented as mean ± SE. ** *p* < 0.01, **** *p* < 0.0001. EST: Ehrich solid tumor.

**Figure 8 foods-12-04139-f008:**
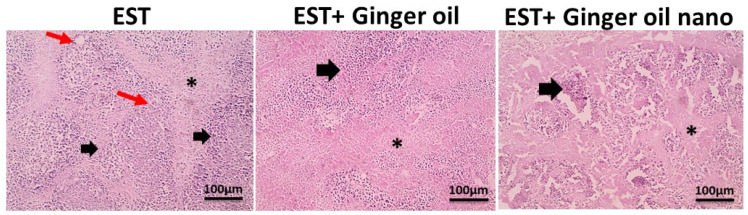
Microscopic pictures of H&E-stained sections from untreated Ehrlich solid tumor (EST) group showing large, round, and polygonal deeply stained tumor cells (thick black arrows), small eosinophilic necrotic zones (*), and newly formed blood capillaries (red arrows). Treated groups demonstrated moderately reduced size of viable areas (thick black arrows), increased size of necrotic areas (*), absence of mitotic figures, and shrunken tumor cells, with the EST + ginger oil nano group showing better improvement. ×100, bar 100 µM.

**Table 1 foods-12-04139-t001:** The dependent and independent formulation variables and their levels according to 3^2 factorial design.

Independent Factors	Low (−1)	Medium (0)	High (1)
X1 = Type of surfactant	Tween 20	Tween 40	Tween 80
X2 = Concentration of surfactant and cosurfactant (%)	10	20	30
**Dependent variables**	**Goal**		
Y1 = Droplet size (nm)	Minimize		
Y2 = PDI	Minimize		
Y3 = ZP (mV)	Maximize		

**Table 2 foods-12-04139-t002:** The composition of different formulations of ginger oil nanoemulsion.

F No.	Ginger Oil % (*v*/*v*)	Type of Surfactant	Surfactant % (*v*/*v*)	Cosurfactant % (*v*/*v*)	Water % (*v*/*v*)
F1	20	Tween 20	5	5	70
F2	20	Tween 20	10	10	60
F3	20	Tween 20	15	15	50
F4	20	Tween 40	5	5	70
F5	20	Tween 40	10	10	60
F6	20	Tween 40	15	15	50
F7	20	Tween 80	5	5	70
F8	20	Tween 80	10	10	60
F9	20	Tween 80	15	15	50

**Table 3 foods-12-04139-t003:** Evaluation of the particle, PDI, and zeta potential of ginger oil nanoemulsion formulations.

F No.	Droplet Size (nm)	PDI	Zeta Potential (mV)
F1	357.17 ± 3.62	0.483 ± 0.10	−34.60 ± 0.95
F2	259.83 ± 1.13	0.375 ± 0.04	−27.90 ± 0.20
F3	162.47 ± 0.85	0.297 ± 0.01	−12.33 ± 1.01
F4	298.51 ± 1.62	0.433 ± 0.02	−35.17 ± 0.59
F5	215.43 ± 2.71	0.404 ± 0.02	−28.33 ± 0.46
F6	109.40 ± 3.08	0.353 ± 0.06	−22.13 ± 0.35
F7	223.37 ± 2.96	0.481 ± 0.04	−39.33 ± 0.96
F8	132.75 ± 2.65	0.407 ± 0.05	−31.90 ± 0.61
F9	56.67 ± 3.10	0.340 ± 0.03	−29.20 ± 0.53

Data are presented in the form mean ± SD; F No., formulation number; PDI, polydispersity index; *n* = 3.

**Table 4 foods-12-04139-t004:** Analysis of variance (ANOVA) for droplet size (Y1), PDI (Y2), and ZP (Y3) of the prepared ginger oil nanoemulsion formulations.

Droplet Size (Y1)						
Source	Sum of Squares	df	Mean Square	F-Value	*p*-Value	
Model	73,087.72	4	18,271.93	215.82	<0.0001	Significant
A-Type of surfactant	22,573.57	2	11,286.78	133.32	0.0002	Significant
B-Amount of Surfactant and cosurfactant	50,514.16	2	25,257.08	298.33	<0.0001	Significant
Residual	338.65	4	84.66			
Cor Total	73,426.37	8				
**PDI (Y2)**						
**Source**	**Sum of Squares**	**df**	**Mean Square**	**F-Value**	** *p* ** **-Value**	
Model	0.0285	4	0.0071	9.33	0.0263	Significant
A-Type of surfactant	0.0009	2	0.0004	0.5815	0.6002	Non-Significant
B-Amount of Surfactant and cosurfactant	0.0276	2	0.0138	18.07	0.0099	Significant
Residual	0.0031	4	0.0008			
Cor Total	0.0316	8				
**ZP (Y3)**						
**Source**	**Sum of Squares**	**df**	**Mean Square**	**F-Value**	** *p* ** **-Value**	
Model	454.93	4	113.73	8.07	0.0338	Significant
A-Type of surfactant	110.12	2	55.06	3.90	0.1147	Non-Significant
B-Amount of Surfactant and cosurfactant	344.81	2	172.41	12.23	0.0198	Significant
Residual	56.40	4	14.10			
Cor Total	511.33	8				

**Table 5 foods-12-04139-t005:** The measurements of pH, viscosity, and CI% of the prepared ginger oil nanoemulsion.

F No.	pH	Viscosity (cP)	CI%
F1	6.71 ± 0.02	2.68 ± 0.05	21.24 ± 0.60
F2	6.24 ± 0.04	7.30 ± 0.32	20.56 ± 0.87
F3	6.18 ± 0.03	20.25 ± 0.08	18.01 ± 0.26
F4	6.24 ± 0.04	2.61 ± 0.10	17.89 ± 0.42
F5	5.95 ± 0.02	6.37 ± 0.16	12.50 ± 0.33
F6	5.87 ± 0.03	19.82 ± 0.46	11.64 ± 0.49
F7	6.82 ± 0.02	2.31 ± 0.03	10.72 ± 0.35
F8	6.84 ± 0.02	6.04 ± 0.19	5.59 ± 0.57
F9	6.87 ± 0.02	14.85 ± 0.26	1.61 ± 0.25

Data are presented in the form mean ± SD; F No., formulation number; CI%, creaming index; *n* = 3.

**Table 6 foods-12-04139-t006:** Effect of treatment with free ginger oil or ginger oil nanoemulsion on survival time and life span of mice bearing Ehrlich solid tumor (EST).

Treatments	Mean Survival Time (Days)	% Increase in Life Span
EST	24 ± 1.528	-
EST + Ginger oil	32 ± 2.08	133%
EST + Ginger oil nano	47.33 ± 3.52	147.9%

Data are presented in the form mean ± SE; EST, Ehrlich solid tumor; *n* = 3.

**Table 7 foods-12-04139-t007:** Effect of treatment with free ginger oil or ginger oil nanoemulsion on liver and kidney function tests.

Treatments	ALT (U/L)	AST (U/L)	Creatinine (mg/dL)	BUN (mg/dL)
Normal	91 ± 8.14	219 ± 19.09	0.44 ± 0.03	36 ± 2.83
EST	334.0 ± 12.1	472.3 ± 20.1	1.31 ± 0.08	76.67 ± 3.587
EST + Ginger oil	294 ± 18.25	426.3 ± 14.08	1.09 ± 0.07	68.27 ± 1.8
EST + Ginger oil nano	211.7 ± 31.95	328.0 ± 25.38	0.73 ± 0.057	57.37 ± 4.43

Data are presented in the form mean ± SE; EST, Ehrlich solid tumor; ALT, alanine aminotransferase; AST, aspartate aminotransferase; BUN, blood urea nitrogen; *n* = 3.

## Data Availability

Data are available from the corresponding author upon request.

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
