# Peer review of "Ginger Oil Nanoemulsion Formulation Augments Its Antiproliferative Effect in Ehrlich Solid Tumor Model"

_foods, 2023, doi:10.3390/foods12224139_

Round 1

Reviewer 1 Report

Comments and Suggestions for Authors

The authors of this study used a 3^2 full-factor design to optimize the oral formulation of ginger oil nanoemulsion for treating breast cancer in in-vivo subcutaneous models. The study was conducted in a well-mannered way, and the authors investigated the physio-chemical characteristics of the nanoemulsion, including its stability measurements. However, the authors must address a few minor comments. A few typos throughout the article need correction, and some other comments have also been listed.

1.    In the introduction's last paragraph, it is recommended to mention the surfactants and co-surfactants used in the study for easy flow from the beginning.

2.      Line 86, detected with repeated words, could be removed accordingly.

3.    In line 162, a typo was noticed with the viable cell number, and in line 164, a typo was noticed.

4.      Line 167, group V, was meant for group IV?

5.   In section 2.9.1, the animal groups and randomization process are unclear and confusing for the reader. It is suggested to rewrite these parts for clarity. Also, mention the number of mice utilized per group. (After rewriting the section 2.9.1, update accordingly in section 2.9.3)

6.      Line 170: is the formula correct for measuring the tumor size/volume?

7.      The sentence from lines 218 and 219 is unclear; complete the sentence.

8.      Line 432, there is a typo.

9.      Figure 6 required a statistical analysis.

10.  Line 450, does it bar 100 µm?

Author Response

Kindly find the responses in the attached file 

Reviewer 2 Report

Comments and Suggestions for Authors

The paper “Ginger Oil Nanoemulsion Formulation Augments its Antiproliferative Effect in Ehrlich Solid Carcinoma Model” discusses the efficacy of nanoemulsion preparation of Ginger essential oil for oral bioavailability and in vivo anti-cancer efficacy.

Ginger oil nanoemulsion were prepared, characterized by TEM and showed key anti-tumor effect against in o Ehrlich solid carcinoma mice.

The concept of the manuscript is interesting considering the bioavailability of natural products as a limiting factor for usage as drug molecule. Studies focusing on addressing limitations with pharmacokinetic properties and ADMET profiles aims to solve key concerns in the development of natural products as herbal medicines.

Some specific ciomments

Abstract

Line 31- Ehrlich solid carcinoma (ESC) model while in line 37, EST bearing mice?? Please revise for clarity

What was the rationale behind investigation anti-breast cancer activity of ginger oil nanoemulsion? Why not any other form of cancer. It is possible that ginger oil nanoemulsion may be more effective against other cancer. Discuss.

Figure 1 and Figure 2 captions- Explain in some details. The details should be discussed in the figure caption, revise likewise for all figures.

Are there any studies reporting the anticancer effect of ginger oil? Discuss by giving examples.

Ginger oil is a natural product extracted from Zingiber officinale. Is it 100% safe to be used as drug for cancer treatment, are there any reports of safety profiles, mention and discuss in the introduction.

Discuss the prospects and limitations associated with ginger oil for the treatment of cancers in the conclusion section.

What are the future goals of the study?

 Minor comments:

Line 62: Zingiber officinale should be in italics, likewise revise all scientific names.

Please revise the manuscript for English language, grammatical errors etc. for clarity and consistency.

References can be improved.

The manuscript may be considered subject to pointwise revision of the suggested comments.

Comments on the Quality of English Language

Moderate English revision is required.

Author Response

(The authors gave the same response as above.)

Reviewer 3 Report

Comments and Suggestions for Authors

The work entitled “Ginger Oil Nanoemulsion Formulation Augments its Antiproliferative Effect in Ehrlich Solid Carcinoma Model” is interesting. However, this work should be significantly improved (corrected), especially the methodological and results parts. Introduction should be rewritten, focusing on background literature and aim of presented work. Materials and methods should be completed with more descriptions and details about methods and used apparats.

 Reviewer's suggestions:

Regarding Introduction

There is no justification for study topic of the work, and a clearly formulated purpose of the manuscript.

Regarding Materias and mehthods

In the work are many editorial errors, e.g.: "Materials" part.

Details about devices -  are missing, e.g. centrifuge, homogenizer. What type, country of manufacturer?

Regarding Materials: The chapter concerns the raw material, but not the test material. The chapter schould be corrected.

Moreover, the characteristic of ginger oil and “Tween 20, 40, 80” are needed.

It is not clear, what the differences are between Tween 20, 40 and 80

Regarding “2.2. Design and preparation of ginger oil nanoemulsion using 3^2 full factorial design”

A very high concentration, of surfactant and cosurfactant (10, 20, and 30%) was used in the work... What was the purpose to using so much amounts of these additives into the emulsion? Please explain.

L 92:” The experiment was designed using the Design expert software version 11.- More details is needed.

 Regarding “2.6.1.. Heating Cooling Cycle (accelerated stability study)”:

Emulsion was subjected to accelerated conditions (4 °C and 45 °C for 6 cycles).  Please gave more details of the examination

Regarding  Statistical analysis:

There is lack of information, about used statistical software.

 Regarding Results

 L.202-207: This is information typical of the work methodology and should be removed from this chapter.

L.284: „In a study conducted by Acevedo-Fani et al. (2015),…” - ??

Regarding Tables and Figures

Table 3, There is no statistical analysis. The results of the statistical analysis should be added. należy dodać wyniki analizy statystycznej.

Figures 1, 2, 3 show the same results as table 3. The graphs should be deleted.

 Figure 5.: The figure  5 show image of one emulsion, it is not known which one?

Tables and Figure 5, 7,8: Explanations of the abbreviations used in the graphs, should be provided below the drawings.

Author Response

(The authors gave the same response as above.)

Round 2

Reviewer 3 Report

Comments and Suggestions for Authors

Dear Authors

The manuscript in its current form is suitable for publication in Foods.